



# Global analysis of the controls on seawater dimethylsulfide spatial variability

George Manville[1], Thomas G. Bell[2], Jane P. Mulcahy[3], Rafel Simó[4], Martí Galí[4,5], Anoop S. Mahajan[6], Shrivardhan Hulswar[6], Paul R. Halloran[1]

[1]Faculty of Environment, Science and Economy, University of Exeter, Exeter, EX4 4PY, UK
     [2]Plymouth Marine Laboratory (PML), Plymouth, PL1 3DH, UK
     [3]Met Office, Exeter, EX1 3PB, UK
     [4]Institut de Ciències del Mar (ICM-CSIC), Barcelona, 08003, Catalonia, Spain
     [5]Barcelona Supercomputing Center (BSC-CNS), Barcelona, 08034, Catalonia, Spain
[6]Indian Institute of Tropical Meteorology (IITM), Ministry of Earth Sciences, Pune, 411008, India

*Correspondence to:* George Manville (gm441@exeter.ac.uk) and Thomas G. Bell (tbe@pml.ac.uk)

**Abstract.** Dimethylsulfide (DMS) emitted from the ocean makes a significant global contribution to natural marine aerosol and cloud condensation nuclei, and therefore our planet's climate. Oceanic DMS concentrations show large spatiotemporal variability, but observations are sparse, so products describing global DMS distribution rely on interpolation or modelling.

Understanding the mechanisms driving DMS variability, especially at local scales, is required to reduce uncertainty in large scale DMS estimates. We present a study of mesoscale and sub-mesoscale (<100 km) seawater DMS variability that takes advantage of the recent expansion in high frequency seawater DMS observations and uses all available data to investigate the typical distances over which DMS varies in all major ocean basins. These DMS spatial variability lengthscales (VLS) are uncorrelated with DMS concentrations. DMS concentrations and VLS can therefore be used separately to help identify

mechanisms underpinning DMS variability. When data are grouped by sampling campaigns, almost 80% of the DMS VLS can be explained using the VLS of sea surface height anomalies, density, and chlorophyll-*a*. Our global analysis suggests that both physical and biogeochemical processes play an equally important role in controlling DMS variability, in contrast with previous results based on data from the low–mid latitudes. The explanatory power of sea surface height anomalies indicates the importance of mesoscale eddies in driving DMS variability, previously unrecognised at a global scale and in agreement

with recent regional studies. DMS VLS differs regionally, including surprisingly high frequency variability in low latitude waters. Our results independently confirm that relationships used in the literature to parameterise DMS at large scales appear to be considering the right variables. However, contrasts in regional DMS VLS highlight that important driving mechanisms remain elusive. The role of sub-mesoscale features should be resolved or accounted for in DMS process models and parameterisations. Future attempts to map DMS distributions should consider the length scale of variability.



## 1 Introduction

Dimethylsulfide (DMS) is a volatile sulfur gas produced by surface ocean microbial food webs and emitted to the atmosphere (Bates et al., 1992). DMS emissions dominate atmospheric biogenic sulfur and form a significant component of natural marine aerosol (Sanchez et al., 2018; Simó, 2001). Aerosols increase light scattering and modify cloud optical properties, thereby contributing to a radiative forcing of climate (Carslaw et al., 2013; Charlson et al., 1987; Galí et al., 2021). The amount, composition, and distribution of natural aerosol in the atmosphere determines the indirect radiative forcing effect of anthropogenic aerosol on climate but is poorly constrained by global climate models (Carslaw et al., 2013). DMS derived sulfate aerosols are ephemeral (~1 day residence time Boucher et al., 2003) and of greater consequence for cloud modulation in remote pristine regions (Halloran et al., 2010). Accurately representing the distribution of natural marine aerosol sources at the resolution necessary to capture the frequency and magnitude of their variability is critical to reducing large uncertainties in the impact of natural aerosol-cloud interactions.

Oceanic DMS production and consumption pathways are complex and the controls of DMS spatial distribution in the global ocean are not fully resolved (Galí & Simó, 2015). The global surface seawater DMS database contains measurements that show large scale temporal and spatial variability in DMS concentrations (Hulswar et al., 2022; Lana et al., 2011). *In-situ* DMS measurements are relatively sparse and limited with respect to global distribution, coverage, and spatiotemporal sampling frequency, which renders the majority of DMS observations insufficient to resolve local and sub-mesoscale variability (Belviso, Moulin, et al., 2004; Lana et al., 2011; Tortell et al., 2011). DMS sampling is globally biased towards spring-summer months (see Fig. S1, Supplementary Material) and has disproportionally targeted biologically productive areas (e.g., northeast Pacific and northwest Atlantic, see Fig. 1), which can lead to an overrepresentation of high DMS concentrations (Galí et al., 2018). Monthly and repeat interannual DMS measurements are rare, and generally restricted to DMS productive areas (Galí et al., 2018). Sparse, infrequent, and seasonally/spatially biased observations of highly variable DMS concentrations create uncertainty because it is hard to quantify the representativeness of the measurements. Sampling uncertainties inevitably propagate through to DMS concentration and flux climatologies, parameterisations, and model outputs (Belviso, Bopp, et al., 2004).

Relatively simple extrapolation methods have been used to fill the gaps between sparse observations to provide globally representative estimates of DMS (Hulswar et al., 2022; Kettle et al., 1999; Lana et al., 2011). More complex algorithms have been generated at the basin or global scale using parameters such as chlorophyll, light, nutrients, surface temperature, and mixed layer depth (Anderson et al., 2001; Aranami & Tsunogai, 2004; Aumont et al., 2002; Belviso, Moulin, et al., 2004; Chu et al., 2003; Galí et al., 2015, 2018; Halloran et al., 2010; Miles et al., 2009; Simó & Dachs, 2002; Vallina & Simó, 2007). Recently a new climatology has been generated using an artificial neural network approach (Wang et al. 2020). The variation in different climatological DMS estimates highlights that the scientific community needs to better understand and map the

processes controlling its oceanic distribution (Belviso, Bopp, et al., 2004; Halloran et al., 2010). Modelled seasonal/regional aerosol-cloud interactions and radiative forcing are directly sensitive to the accuracy/choice of seawater DMS estimates

(Mahajan et al., 2015; Woodhouse et al., 2010, 2013).

Few studies have focussed on local and sub-mesoscale DMS variability. This study explores the processes that appear to govern DMS variability at the <100 km scale and investigates whether these align with the variables used within large scale DMS parameterisations. An improved understanding of sub-mesoscale DMS variability will aid the development of future

climatological flux estimates and the appropriate radius of influence that sparse observations should be afforded when smoothing and interpolating in situ observations.

Variability lengthscale (VLS) analysis is a powerful tool for quantifying sub-mesoscale variability. VLS analysis can be used to indicate the lowest sampling resolution necessary to capture most of the spatial variability (Royer et al., 2015). High

resolution measurements are required to assess small scale variability. For example, to observe variations within 10 km when the research ship is travelling at $8$ m s$^{-1}$ requires measurements every 20 mins. Instruments that can observe variability at these high resolutions have been deployed in recent years and have contributed substantially to the global DMS database (Hulswar et al., 2022). A wealth of high frequency DMS data offers a new opportunity for a global analysis of the drivers of DMS variability at small scales.


VLS analysis for DMS has been applied in only a few studies, with most focusing on a specific region and/or a single sampling campaign (e.g., Ross Sea; (Tortell et al., 2011; Tortell & Long, 2009), northeast subarctic Pacific (Asher et al., 2011; Nemcek et al., 2008; Tortell, 2005)). A larger scale VLS analysis was undertaken on the 7-month low–mid latitude global circumnavigation conducted during the Malaspina Expedition 2010 (Royer et al., 2015). Royer et al. (2015) combined their

VLS analysis with VLS values from 3 high latitude studies (7–15 km, Asher et al., 2011; Nemcek et al., 2008; Tortell et al., 2011) and reported an inverse relationship between DMS VLS and latitude (R = –0.74, p < 0.005). Royer et al. (2015) also reported that biological variables dominate over physical variables as drivers of DMS VLS in low latitude regions. While it is tempting to draw global conclusions from the similarities and differences between these studies, each study adopts a slightly different approach to the data treatment, measurement of interpolation error, and/or classification of VLS (see Table S1,

Supplementary Material).

This study applies a single, objective VLS analysis to high frequency global DMS observations over the past 15 years (Fig. 1 & S1). The dataset used includes all available data from previous VLS studies. Our study assesses whether the factors controlling DMS variability can be identified using a sub-mesoscale variability analysis across all ocean basins. Sect. 2

describes the datasets used and the VLS methodology. Sect. 3 presents results including global VLS statistics, regional patterns





of DMS variability, and drivers of DMS variability. Finally, the findings are discussed in Sect. 4, with conclusions made in Sect. 5.

## 2 Data & methods

### 2.1 Seawater DMS data

The majority of DMS data are sourced from the global surface seawater DMS database (GSSDD; see https://saga.pmel.noaa.gov/dms/). Selection criteria are used to identify datasets suitable for sub-mesoscale VLS analysis: a minimum of 100 data points in total and ≤1 hour between measurements. Applying these filters results in 37 eligible datasets (collected between 2004 and 2019). The filters broadly separate the DMS database by sampling method, highlighting the rapid shift during the early 2000's from discrete, low frequency gas chromatography analytical systems, to continuous, semi-
automated high frequency mass spectrometry (Bell et al., 2012). Additional data are from the Malaspina Expedition in 2010-2011 (M10, Royer et al., 2015), the North Atlantic Aerosol and Marine Ecosystem Study in 2015–2018 (NAAMES; Bell et al., 2021; Fig. 1 & S1, Table S2, campaign numbers: 33 (blue), 34 (green), 35 (red), 36 (yellow)), and the Southern oCean SeAsonaL Experiment in 2019 (SCALE; Manville et al. In Prep; Fig. 1 & S1, Table S2, campaign number: 37 (green)). The M10 circumnavigation data are split spatiotemporally into 3 datasets, each broadly covering different ocean basins (Fig. 1 &
S1, Table S2, campaign numbers: 30 (M10a, black), 31 (M10b, dark red), 32 (M10c, cyan)).

### 2.2 Ancillary *in-situ* & coincident satellite measurements

Ancillary *in-situ* and remotely sensed data are used to explore the processes that may be driving DMS variability. *In-situ* sea surface salinity (hereafter salinity) and temperature (SST) from each DMS dataset are used to derive sea surface density (hereafter density) (see Fernandes, 2014).


Satellite chlorophyll-*a* (Chl) and sea surface height anomaly (SSHA) data are extracted along the coordinates of each cruise track using the NASA SeaDAS software (version 7.5.3). NASA MEaSUREs L4 0.17° 5-day SSHA are used to explore the role of eddies in driving DMS variability (Zlotnicki et al., 2019). NASA MODIS-Aqua L3 4 km monthly Chl is used as a proxy for plankton biomass and biological productivity (NASA Goddard Space Flight Center, 2018).

### 2.3 Data processing

Underway data are screened to only include data acquired when ship speed was >1 m s$^{-1}$, to avoid measurements made when ships were sampling on station. Ship speed is calculated from distance and time between measurements. Each DMS dataset and all its ancillary data is divided into transects. Transects are defined as continuous data sections with a minimum sampling frequency of 1 hour, which excludes all data with a spatial resolution >30 km. The vast majority (83%) of observations captured
by the temporal filter are <2.2 km apart. The minimum transect length is calculated in two stages: 1) the linear distance between





the start and end of a continuous data section must be >100 km to avoid campaigns that targeted a specific area multiple times (e.g., a productive bloom or mesoscale eddy); 2) each dataset is divided into equal length transects, with an along track distance of at least 100 km. The initial data processing yields 1039 continuous transects from 37 DMS campaigns, with each transect 100–199 km in cumulative length (Fig. 1).

**2.4 Variability lengthscale (VLS) analysis**

Previous DMS VLS studies have not applied a standardised or consistent approach (Asher et al., 2011; Nemcek et al., 2008; Royer et al., 2015; Tortell, 2005; Tortell et al., 2011; Tortell & Long, 2009). The analysis presented here adopts the method used to study the VLS of seawater $CO_2$ (Hales & Takahashi, 2004), which was later applied to DMS by Tortell et al. (2011) and Nemcek et al. (2008).


The highest observational DMS sampling resolution in the datasets is typically between 0.2 and 2.2 km. Each data transect is subsampled starting from the first data point, at increasingly coarse spacings ranging from 2.2 km to half the length of the transect (the lowest possible resolution), increasing in 0.2 km increments. At each subsampling resolution, the first and last subsampled points of the data transect define the subsampling window. Subsampled data across the subsampling window are

linearly interpolated to the resolution of the original data. Where the subsampling window matches the length of the data transect, the interpolation error associated with the subsampling resolution is calculated as the root mean squared error (RMSE) between the original and the interpolated values. Where the subsampling window is not equal to the length of the transect, the window is shifted along the transect, incrementing by one data point, and the transect is re-subsampled. Re-subsampled data are linearly interpolated across the shifted window, and the RMSE is re-calculated. The subsampling window is repeatedly

shifted along the data transect and interpolation RMSE re-calculated until the subsampling ends on last data point of the transect. The error associated with the subsampling resolution is taken as the average of all the RMSE values produced by sliding the window across the data transect at that resolution. RMSE is calculated following Eq. (1):

$$RMSE = \sqrt{\overline{(Obs - Interp)^2}} \tag{1}$$

RMSE typically increases in proportion to the coarseness of the subsampling until a maximum error plateau, or asymptote, is

reached. The maximum error plateau corresponds to the total variance of the dataset (Belviso et al., 2004; Tortell et al., 2011). The trend in RMSE as a function of subsampling resolution is well described by a non-linear first-order inverse exponential rise function following Eq. (2):

$$E_x = E_\infty \left(1 - e^{\left(-\frac{x}{VLS}\right)}\right) \tag{2}$$

where $E_x$ is the interpolation error at subsampling resolution $x$, $E_\infty$ is the asymptotic maximum interpolation error at an infinite

subsampling resolution, and VLS is the characteristic lengthscale of variability. VLS is determined by the sub-sampling





resolution (interpolation distance) where a tangent of the initial slope intersects with the maximum error ($E_\infty$, Fig. 2). VLS also corresponds to the intersect on the curve ($E_x$) that is 63% of $E_\infty$, i.e., Eq. (3):

$$\frac{E_x}{E_\infty} = 1 - e^{\left(-\frac{x}{\text{VLS}}\right)} \approx 0.63 \tag{3}$$

Previous work suggested that a sudden change (or 'breakpoint') in the RMSE slope can be used to characterise the DMS VLS
(Asher et al., 2011; Royer et al., 2015). However, this approach is unreliable, because the data assessed in this study shows that the breakpoint does not always occur, and its identification is subjective (see Table S1, Supplementary Material).

An inverse exponential rise function (Eq. 2&3) is used here to objectively derive VLS. The objective VLS method is applied to all 1039 transects and six parameters: DMS, SST, salinity, density, Chl, SSHA.

**2.4.1 Quality assurance and VLS statistics**

Two filters are used to identify viable data transects. VLS is rejected if the distance is greater than the maximum subsampling / interpolation distance (equal to half the transect length), which only occurred in very noisy datasets. The second filter is the quality of fit to the data using the residual standard error (RSE) (Fig. 2b), which is defined as RSE $= \sqrt{(ss_{res}/n)}$ where $n$ is the number of data points in the transect, and $ss_{res}$ is the sum of the squares of the residuals, i.e., $ss_{res} = \sum(RMSE - curve)^2$.

The RSE is normalised using the maximum RSE of the curve (i.e., (RSE/RSE at the asymptote) × 100) and if the normalised RSE exceeds 10%, the curve is deemed to inadequately describe the data and the transect is rejected. The two quality control filters reduce the initial 1039 transects to 763 'viable' transects.

The distributions of VLS from the 763 transects are skewed for all parameters (Fig. 3 & S2). The geometric mean and geometric standard deviation (GSD) are computed to assess central tendency and spread while accounting for skew in the data. Note that the geometric mean is regularly referred to as the 'average' within this manuscript to aid readability. Transects are grouped and averaged by sampling campaign to assess underlying spatial and temporal (regional and seasonal) patterns of variability. Average VLS distances are calculated for each sampling campaign and for all parameters (VLS$_{\text{DMS}}$, VLS$_{\text{SST}}$, VLS$_{\text{salinity}}$,
VLS$_{\text{density}}$, VLS$_{\text{Chl}}$, VLS$_{\text{SSHA}}$). A minimum threshold of four transects was necessary before calculating a campaign average VLS. Exclusion of campaigns with <4 transects reduced the total number of campaigns from 37 to 35.

Correlation and multiple linear regression (MLR) are used to explore the global controls on VLS$_{\text{DMS}}$ (see Sect. 3.3.1 and 3.3.2; Table 1). The campaign average VLS used in each correlation and regression analysis only includes transects where coincident
VLS can be calculated from the DMS and non-DMS parameters. MLR models with two input parameters contain 20-26 datasets, and MLR models with three input parameters contain 11-15 datasets (see Table 1). The relative importance of the





input parameters in each MLR model are calculated based on the incremental $R^2$ used to determine interactional dominance (defined as the incremental $R^2$ contribution of each predictor to the complete model; Azen & Budescu, 2003).

## 3 Results

### 3.1 Global VLS statistics

The global average DMS concentration and geometric standard deviation (GSD) for the viable transects covered in this study are 2.23 nM (average) and 2.29 nM (GSD), which is similar to the global average and GSD from the GSSDD (2.66 nM (average), 2.88 nM (GSD); date of last access 15 April 2022). The similarity between the two datasets suggests that the data used in this study is representative of global observations. The global average and GSD of $VLS_{DMS}$ from all 763 transects are 12.57 km and 2.33 km, respectively (Fig. 3). Global average $VLS_{DMS}$ is the smallest of the six parameters tested, with $VLS_{SSHA}$ the most similar (15.76 km, 1.77 km GSD) (Fig. 3). Global average $VLS_{Chl}$ is slightly larger (20.89 km, 1.67 km GSD) and similar to global average $VLS_{density}$ (20.21 km, 1.76 km GSD), and its components $VLS_{SST}$ (21.23 km, 1.73 km GSD) and $VLS_{salinity}$ (19.52 km, 1.84 km GSD) (Fig. 3 & S1).

All six parameters have an average spatial variability that is tens of kilometres in all regions. Global average $VLS_{SSHA}$ is similar (within 4 km) to global average $VLS_{DMS}$. Global average VLS for all other parameters are within 9 km of global average $VLS_{DMS}$. Campaign average $VLS_{DMS}$ ranges from 2 to 30 km, which is the same order of magnitude as the range of 7–50 km reported by other DMS variability studies (Asher et al., 2011; Nemcek et al., 2008; Royer et al., 2015; Tortell, 2005; Tortell et al., 2011). Note that a detailed comparison between studies should be treated with caution because each have used different methods to identify the VLS.

### 3.2 Regional patterns of DMS variability

The Southern Hemisphere subtropical gyres are permanently stratified biomes (Fay & McKinley, 2014), and these regions have consistently small $VLS_{DMS}$ (Fig. 4). The relative homogeneity of $VLS_{DMS}$ in some oligotrophic domains is not replicated in the VLS of any other parameters (Fig. S3). The average (22.06 km) and GSD (1.56 km) of $VLS_{DMS}$ in the Peruvian upwelling (East equatorial Pacific) is consistently larger than the global average (12.57 km, 2.33 km GSD) (Fig. 4). $VLS_{DMS}$ is small in the sub-tropical Pacific and South Atlantic, (Fig. 4). Larger $VLS_{DMS}$ is found along parts of the Pacific and Atlantic coastlines of North America, with smaller $VLS_{DMS}$ further offshore (Fig. 4, inset).

$VLS_{DMS}$ in the Arctic, northeast Pacific, northwest Atlantic, and SE Indian open ocean regions is highly variable. The Southern Ocean has $VLS_{DMS}$ generally below the global average and features some localised pockets with larger VLS (Fig. 4). DMS concentration variability in mid–high latitude regions is seasonal (Hulswar et al., 2022) and $VLS_{DMS}$ could be influenced by the season / time of year.





### 3.3 Drivers of DMS variability

#### 3.3.1 Transect and campaign average VLS regressions

Simple linear regressions are used to explore the relationship between $VLS_{DMS}$ and VLS for SST, salinity, density, Chl and

SSHA. The possibility of a relationship with latitude (as discussed in Royer et al. (2015)) is also investigated. Transect and

campaign average $VLS_{DMS}$ do not vary with latitude ($R^2 = 0.02$, n = 35, $p > 0.05$; Table 1). No significant relationships are

observed between transect $VLS_{DMS}$ and VLS for SST, SSS, density, Chl and SSHA. Averaging transect VLS data into

campaign averages reduces the noise and enables statistically significant relationships to be identified. Campaign average

$VLS_{density}$ explains 37% of the variations in $VLS_{DMS}$ (Table 1; Fig. 5a). $VLS_{SSHA}$ (used as an indicator of the dynamic eddy

field in the open ocean) and $VLS_{Chl}$ each explain approximately half of the campaign average $VLS_{DMS}$ (46% and 47%,

respectively; Table 1; Fig. 5b&c).

#### 3.3.2 Multiple linear regression of $VLS_{DMS}$

Multiple linear regression (MLR) is used on the campaign average VLS for SST, salinity, density, Chl and SSHA to explore

$VLS_{DMS}$ variance (Table 1; see Table S4, Supplementary Material for regression coefficients). Eleven MLR combinations were

tested, and all results are significant ($p < 0.01$) except for the combination of $VLS_{Chl}$, $VLS_{SSHA}$ and $VLS_{SST}$ (Model 17, Table

1). Note that the number of available datasets is reduced in the MLR models that have more input parameters, which results in

the contribution of fewer data (campaigns) to the result. The number of input data is substantially increased if campaign

averages are calculated without filtering the data prior to correlation so it only contains data where the two correlated

parameters are co-located. Relaxing the criteria such that the transects need not be coincident increases the number of

campaigns that can be included in each MLR model. The 'relaxed criterion' approach is less robust but gives similar results to

those presented here (see Table S3, Supplementary Material).

Individual $VLS_{Chl}$ and/or $VLS_{SSHA}$ regressions with $VLS_{DMS}$ are outperformed (i.e., $R^2 > 0.47$) by four MLR combinations

(Models 7-10, Table 1). The combination of $VLS_{density}$ and $VLS_{Chl}$ (Model 9, Table 1) substantially improves the regression

with $VLS_{DMS}$ (adjusted $R^2$ increases to 0.63). MLR Model 9 has the most campaigns (n = 26) of any model, and the third

highest number of available data transects (n = 224). $VLS_{Chl}$ (54%) and $VLS_{density}$ (46%) make approximately equal

contributions to the changes in $VLS_{DMS}$ described by Model 9.

The largest amount of $VLS_{DMS}$ variability explained by the MLR models uses the combination of $VLS_{SSHA}$, $VLS_{Chl}$ and

$VLS_{density}$, improving the adjusted $R^2$ to 0.77 (Model 7, Table 1). $VLS_{SSHA}$ is the dominant parameter in Model 7 (52% of the

explained variance), with $VLS_{Chl}$ and $VLS_{density}$ accounting for 34% and 14%, respectively. Combining $VLS_{Chl}$ and $VLS_{SSHA}$

(MLR Model 11) reduces the available input data (n = 20) and does not increase the explained variance in $VLS_{DMS}$ compared

to using only one or other of the input parameters. $VLS_{SSHA}$ and $VLS_{Chl}$ dominate the explained variance in MLR models when

paired with one other variable (Models 12–16, Table 1).





## 4 Discussion

### 4.1 Global statistics

This is the first study of sub-mesoscale seawater DMS variability from a global perspective. Spatial variability lengthscale analysis is applied to every ocean basin, and at different times of year, using a consistent methodology. Characteristic spatial variability in all six parameters (DMS, SST, salinity, density, Chl, SSHA) occurs at the low mesoscale (in the tens of kilometres) in all regions. Campaign average $VLS_{DMS}$ ranges from 2-30 km (Table S2, Supplementary Material), in general agreement with previous work (Asher et al., 2011; Nemcek et al., 2008; Royer et al., 2015; Tortell, 2005; Tortell et al., 2011; Tortell & Long, 2009). There is no correlation between campaign average DMS concentration and $VLS_{DMS}$ ($R^2 = 0.01$, $p > 0.05$), which suggests that understanding the variability may be a helpful and independent approach to understanding the processes that control surface ocean DMS.

### 4.2 Regional patterns of DMS variability

$VLS_{DMS}$ is broadly above average at the edge of ocean basins e.g., parts of northwest Atlantic, northeast Pacific, the California coast (Fig. 4, inset). It may be possible that some coastal DMS invariability is driven by large phytoplankton blooms, which previous local/regional studies suggest can dominate coastal domains (Asher et al., 2011; Nemcek et al., 2008). This work does not investigate the detail of drivers of DMS variability in individual regions or domains.

Open ocean domains such as the sub-tropical gyres in the Southern Hemisphere have consistently small $VLS_{DMS}$, a feature not evident in the VLS of the other parameters (Fig. 4 & S2). Short lengthscales of DMS variability in stable stratified biomes offers the opportunity for future work to re-examine these regions for as yet unidentified drivers of variability. Most low latitude DMS data used in this study originate from a single sampling campaign (e.g., Malaspina Expedition 2010; Royer et al., 2015). To test if small $VLS_{DMS}$ is a persistent feature in under-sampled sub-tropical open oceans, more high-resolution observations are needed.

Factors driving temporal DMS variability are not explored in this study. However, complex $VLS_{DMS}$ fluctuations at high latitudes (e.g., northwest Atlantic, northeast Pacific, Southern Ocean; Fig. 4) may be capturing variations in both space and time. $VLS_{DMS}$ in high latitude dynamic regions could be related to the seasonality of biological productivity and eddy activity (see Asher et al., 2011; Behrenfeld et al., 2019; Bell et al., 2021; Fox et al., 2020; Gaube et al., 2019; Lana et al., 2011; McGillicuddy, 2016). Additionally, it is plausible that $VLS_{DMS}$ in the polar regions may be sensitive to the seasonal impact of sea ice on biogeochemical processes (see Galí et al., 2021; Lannuzel et al., 2020; Stefels et al., 2018). There are not enough repeat measurements made in high latitude (high seasonal variability) regions to establish the impact of seasonality on $VLS_{DMS}$. In this study, the only region sampled during different seasons is the northwest Atlantic (4 Atlantic NAAMES campaigns; (Bell et al., 2021) and there is not yet compelling evidence of a temporal difference between the $VLS_{DMS}$ of these





cruises/seasons. $VLS_{DMS}$ of the NAAMES 1 transects (November; average = 11.93 km, GSD = 1.76 km) are significantly different ($p < 0.01$) from the transect $VLS_{DMS}$ of NAAMES 3 (t = –3.38 , $p < 0.01$; September; average = 20.89 km, GSD = 1.69 km) and NAAMES 4 (t = –3.31, $p < 0.01$; March/April; average = 21.94 km, GSD = 1.57 km), but not from NAAMES 2

(t = –2.22, $p = 0.03$; May/June; average = 18.4 km, GSD = 1.56 km). $VLS_{DMS}$ of the NAAMES 2, 3 & 4 transects are not significantly different from each other (all $p > 0.3$).

## 4.3 Drivers of DMS variability

The variance in campaign average $VLS_{DMS}$ data explained by physical processes (represented by $VLS_{SSHA}$) is as important as biogeochemical processes (represented by $VLS_{Chl}$), with each parameter able to explain just under half of the $VLS_{DMS}$ (Models

1&2, Table 1; Fig. 5). This conclusion contrasts with the findings of Royer et al. (2015) who find in the low–mid latitudes the majority of $VLS_{DMS}$ (65%) is more similar to the VLS of biological variables that represent biomass and physiology (Chl and fluorescence) than to the VLS of physical variables. These contrasting conclusions potentially reflect the fact that length scales of physical oceanographic variability increase towards the equator (Jacobs et al., 2001).

A larger proportion of campaign average $VLS_{DMS}$ variability (77%) can be explained using $VLS_{Chl}$, $VLS_{SSHA}$ and $VLS_{density}$ (Model 7, Table 1) compared to just $VLS_{SSHA}$ or $VLS_{Chl}$. The data included in the $VLS_{SSHA-Chl-Density}$ MLR (Model 7, Table 1) is a subset but includes at least one campaign from each major ocean basin (Fig. S4, Supplementary Material) and is thus a significant relationship with global applicability. $VLS_{SSHA}$ explains the majority of $VLS_{DMS}$ in the $VLS_{SSHA-Chl-Density}$ MLR (52%) (Model 7, Table 1) and improves the prediction of changes in $VLS_{DMS}$ compared to using just $VLS_{Chl}$ and $VLS_{density}$

(Model 9, Table 1). The $VLS_{SSHA-Chl-Density}$ MLR (Model 7, Table 1) includes measurements from the NAAMES4 (2018) cruise, which targeted a substantive eddy and observed a persistent high Chl feature coincident with elevated DMS levels (Bell et al., 2021). The water mass within an eddy tends to be retained by the circulation, such that plankton within the eddy are accumulated under relatively stable physics (upwelling or downwelling) and consistent biogeochemical conditions (Bell et al., 2021). Eddies may thus drive conditions where DMS variability is closely associated with biological activity and a clear co-

variation in VLS is observed, even if the relationship between DMS and Chl concentration is less obvious (della Penna & Gaube, 2019). The relationship between eddy structure, biogeochemistry and DMS may explain the link between changes in $VLS_{DMS}$, $VLS_{SSHA}$, and $VLS_{Chl}$. The importance of $VLS_{SSHA}$ for predicting $VLS_{DMS}$ is consistent with results recently reported by McNabb & Tortell (2022), who apply two independent machine learning techniques to analyse DMS in the northeast Pacific. McNabb & Tortell (2022) demonstrate the power of mesoscale eddies for predicting DMS variability (Spearman

correlation coefficients = 0.35 and 0.42, depending on the machine learning method employed), using the same SSHA product used in this study (using only summertime measurements, 1997 – 2017).



### 4.4 Scale-independent seawater DMS predictability?

SSHA reflects surface mixing and changes to the mixed layer depth (MLD) (Gaube et al., 2019). Low resolution measurements have previously been used to predict regional and seasonal trends in DMS. Several studies have parameterised DMS as a
function of surface mixing, light, and Chl (Anderson et al., 2001; Aranami & Tsunogai, 2004; Aumont et al., 2002; Belviso, Moulin, et al., 2004; Galí et al., 2018; Simó & Dachs, 2002; Vallina & Simó, 2007). For example, Simó & Dachs (2002) use climatological MLD and remotely sensed Chl to estimate average DMS concentrations, while Vallina & Simó (2007) use climatological MLD, surface irradiance and light attenuation to estimate surface DMS from the 'solar radiation dose'. Galí et al. (2018) employ an algorithm driven by climatological Argo MLD and satellite derived Chl, SST, and photosynthetically
active radiation (PAR). DMS parameterisations with global coverage require the powerful tool that remote and autonomous observations provide and do reasonably well at predicting spatially and seasonally averaged surface seawater DMS (e.g., Galí et al., 2018; Simó & Dachs, 2002; Vallina & Simó, 2007). However, some studies have questioned whether such parameterisations are overly reliant on spatial/temporal averaging (e.g., Derevianko et al., 2009). Regional studies have tested empirical predictive relationships for DMS with varying degrees of success (Asher et al., 2011; Royer et al., 2015). The
spatial/temporal averaging used to develop the global parameterisations may lead to an over-confidence in current predictive capabilities because key parameters are not included. This study supports the choice of variables used in existing empirical parameterisations – surface seawater DMS spatial variability in the global ocean is at least partially explained by physical mixing and biological activity. In addition, statistically significant MLR relationships are only obtained once the transect data are averaged by campaign, which suggests that mesoscale DMS variability is determined by processes that are missing from,
or are not captured by, this analysis.

### 4.5 Study limitations and unidentified drivers of DMS variability

This work provides as comprehensive assessment of DMS variability across the global ocean as existing data allow, yet many regions have not yet been sampled at high enough resolution to permit an assessment of $VLS_{DMS}$. For example, only seven of the 37 campaigns in this study have made high resolution DMS measurements in low latitude waters (30ºN–30ºS). There is a
seasonal sampling bias within the DMS database, and the northwest Atlantic is the only region to have been assessed for VLS throughout the seasonal cycle (Bell et al., 2021). More data are needed and would be useful.

Satellite-derived $VLS_{Chl}$ and $VLS_{SSHA}$ have been used to predict $VLS_{DMS}$ (e.g., Model 7, Table 1), but this relies on the assumption that the satellite-retrieved data are representative of phytoplankton productivity and eddy activity throughout the
research cruise/campaign. Satellite retrievals for Chl with higher than monthly temporal resolution or, in the case of SSHA, higher than 0.17º spatial resolution, may improve the ability to explain variance in $VLS_{DMS}$.





Transect lengths between 100 and 199 km are used to ensure comparability between datasets/regions because VLS results from previous studies appear to be sensitive to the length of data transect (see Fig. S5, Supplementary Material). However, by
limiting the transect length, it is difficult to identify large eddies using $VLS_{SSHA}$. Eddy length scales are typically larger at low latitudes due to the dependence of the Coriolis parameter on latitude (Chelton et al., 1998). The maximum VLS in this study is between 50 and 99.5 km (half the transect length), which is long enough to capture the eddy variability at latitudes where the eddy length scale is related to the Rossby radius of deformation, i.e., poleward of 30º where the deformation radius is < 30 km (Eden, 2007). Equatorward of 30º eddy length scales are not well predicted by the Rossby radius of deformation and can
exceed 50 km (Eden, 2007; Klocker et al., 2016; Rhines, 1975; Scott & Wang, 2005; Tulloch et al., 2011). The $VLS_{SSHA}$ analysis approach used in this study is designed to identify the dominant scale of variability in physical features up to 50–99.5 km, therefore it may not capture the full extent of variability associated with large eddies at low latitudes. Large eddies will however still be captured in the $VLS_{SSHA}$ analysis where a transect segments an eddy without passing through its centre. We also note that although SSHA is used to represent eddy features, at the equator stratification and strong westward currents tend
to dominate SSHA variability rather than rotation and eddy transport (Williams & Follows, 2011).

$VLS_{DMS}$ in the subtropical gyres is typically small (<10 km; Fig. 4), which is qualitatively consistent with the short (days) response time of DMS to perturbations in the dynamic equilibrium of DMS production and consumption in these waters (Galí & Simõ, 2015). $VLS_{DMS}$ in subtropical waters does not correspond well with the VLS of any of the other parameters (Fig. S3,
Supplementary Material). Cycling of reduced sulfur compounds in sub-tropical waters is well-documented to be part of a different biogeochemical regime compared to productive, higher latitude waters (e.g., Galí & Simõ, 2015; Toole et al., 2003). In stable oligotrophic regions where there is less variability in physical mixing and phytoplankton productivity, $VLS_{DMS}$ could thus be dominated by alternative parameters that drive variability in the biological cycling of DMS.

The so-called 'summer paradox' describes the seasonal misalignment between maximum concentrations of phytoplankton biomass and DMS in low latitude waters and has been challenging to model (e.g., Galí & Simõ, 2015; Polimene et al., 2012; Toole et al., 2008; Vallina et al., 2008). In these areas, characterized by low seasonal amplitude in phytoplankton biomass, changes in phytoplankton species succession and physiological stress control DMS production yields and rates, and ultimately DMS seasonality. By contrast, aggregated loss processes exhibit low seasonal variability and are insufficient to explain large-
scale DMS seasonality in 'summer paradox' areas (Galí & Simó, 2015). Previous studies observed important short-term variations in the balance between DMS sources and sinks in oligotrophic waters, concomitant with meteorological forcing (Royer et al., 2016). Hence, it is plausible to hypothesize that subtle changes in this balance can explain some of the variance in $VLS_{DMS}$. Light exposure in surface waters influences plankton physiological production and stress, photochemical reactions, and bacterial activity, and thus has a significant impact on the cycling of reduced sulfur in oligotrophic regions (see Toole &
Siegel, 2004; Vallina et al., 2008). These factors have not been included in the present study.

## 5 Conclusions

This study presents a comprehensive and objective analysis of DMS variability based on a large set of high frequency global observations. The work shows that the variability lengthscale for DMS is typically small (< 30 km) and that a substantial proportion of the campaign average variance can be explained by the VLS of key biological (Chl) and physical (density, SSHA) observations (Model 7, Table 1). The results improve confidence in the validity of the biological and physical
parameters used to currently parameterise seawater DMS at large scales and used in many global climate models (e.g., Bock et al., 2021; Galí et al., 2018; Mulcahy et al., 2020; Simó & Dachs, 2002). However, there is substantial variability in $VLS_{DMS}$ when assessing individual transects, which suggests that unaccounted-for variables are also important (e.g., light, wind speed, microbial activity). Making high frequency measurement of these parameters at the same time as high frequency DMS
measurements may help elucidate their role in DMS cycling.

### Data availability

DMS and ancillary *in-situ* data (SST and salinity) are sourced from the global surface seawater DMS database (GSSDD; https://saga.pmel.noaa.gov/dms/; last access: 15 April 2022), and supplied by authors RS and ASM (Malaspina Expedition in 2010-2011, M10), TB (North Atlantic Aerosol and Marine Ecosystem Study in 2015–2018, NAAMES;
https://doi.org/10.5067/SeaBASS/NAAMES/DATA001), and GM (Southern oCean SeAsonaL Experiment in 2019, SCALE). Requests for access to M10 and SCALE DMS data can be sent to the corresponding authors (GM and TB). Satellite data are available in online NASA repositories for chlorophyll-*a* (https://doi.org/10.5067/AQUA/MODIS/L3M/CHL/2018) and sea surface height anomalies (https://doi.org/10.5067/SLREF-CDRV2).

### Supplement

The supplement related to this article is available online at: … [insert DOI]

### Author Contributions

GM, PH, TB and JPM devised the study. GM conducted the analysis and interpretation, and wrote the manuscript, with input from PH and TB. JPM, MG, and RS provided insight and helped improve the analysis and interpretation. All co-authors contributed to the manuscript.

### Competing Interests

The authors declare that they have no conflict of interest.



**Acknowledgements**

This work was supported by the UK Natural Environmental Research Council (through a PhD studentship for GM: NE/R007586/1; and the CARES project for TB: NE/W009277/1). JPM was supported by the Met Office Hadley Centre Climate Programme funded by BEIS and also received funding from the European Union's Horizon 2020 research and innovation programme under grant agreement n°101003536. This research was supported by the Spanish national funding plan for science through project BIOGAPS (CTM2016-81008-R) to RS, through project DMS-Cons (202230I123) to MG, and through the 'Severo Ochoa Centre of Excellence' grant (CEX2019-000928-S) to the ICM-CSIC. RS is a holder of a European Research Council Advanced Grant (ERC-2018-ADG-834162) under the EU's Horizon H2020 research and innovation programme. The Indian Institute of Tropical Meteorology is funded by the Ministry of Earth Sciences, Government of India.

We thank the contributors of high frequency DMS data (Archer, Herr, Jarnikova, Johnson, Marandino, Royer, Saltzman, Tortell, and Zhang) for making their DMS data available (see Table S2, Supplementary Material for full details) via the Global Surface Seawater DMS Database.

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





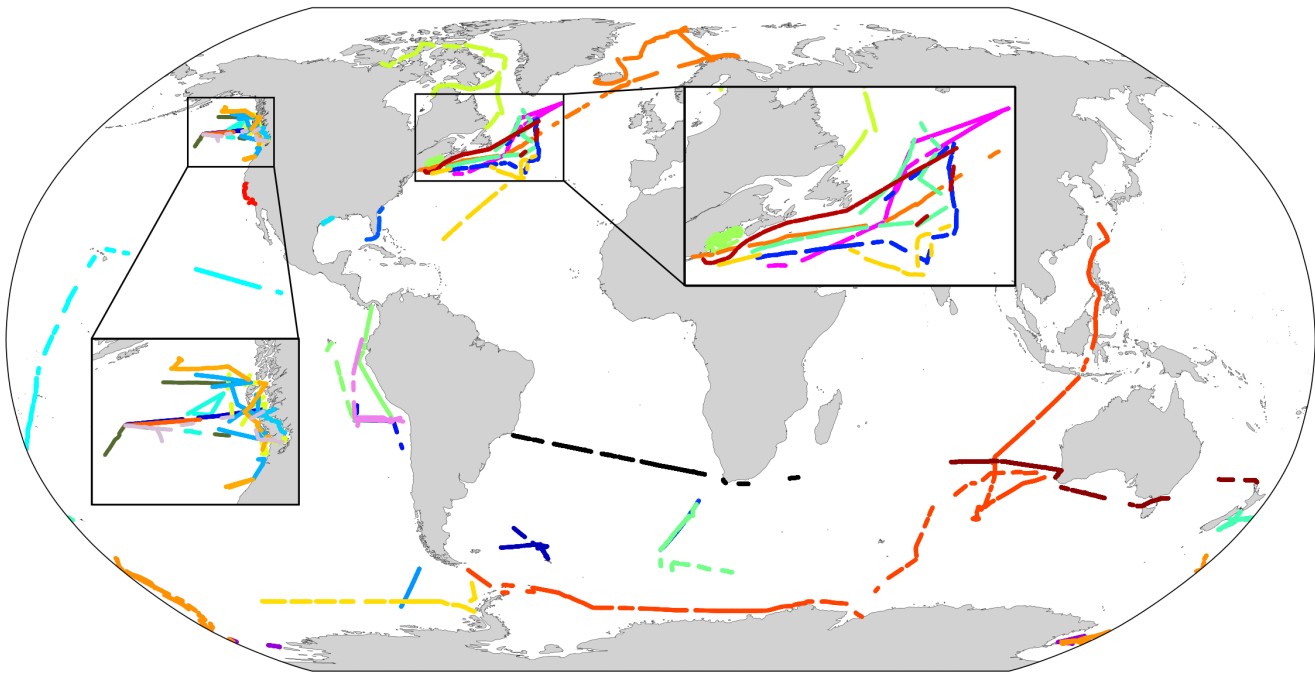

**Figure 1: Global extent of the 37 high frequency DMS campaigns included in this analysis (coloured). Data are only shown for the underway transects used in the VLS analysis (see Sect. 2.3). Insets show detail for northeast Pacific and northwest Atlantic regions**
**with multiple sampling campaigns (see Table S2 and Fig. S1 in the Supplementary Material for metadata relating to each sampling campaign and the spatiotemporal distribution, respectively).**





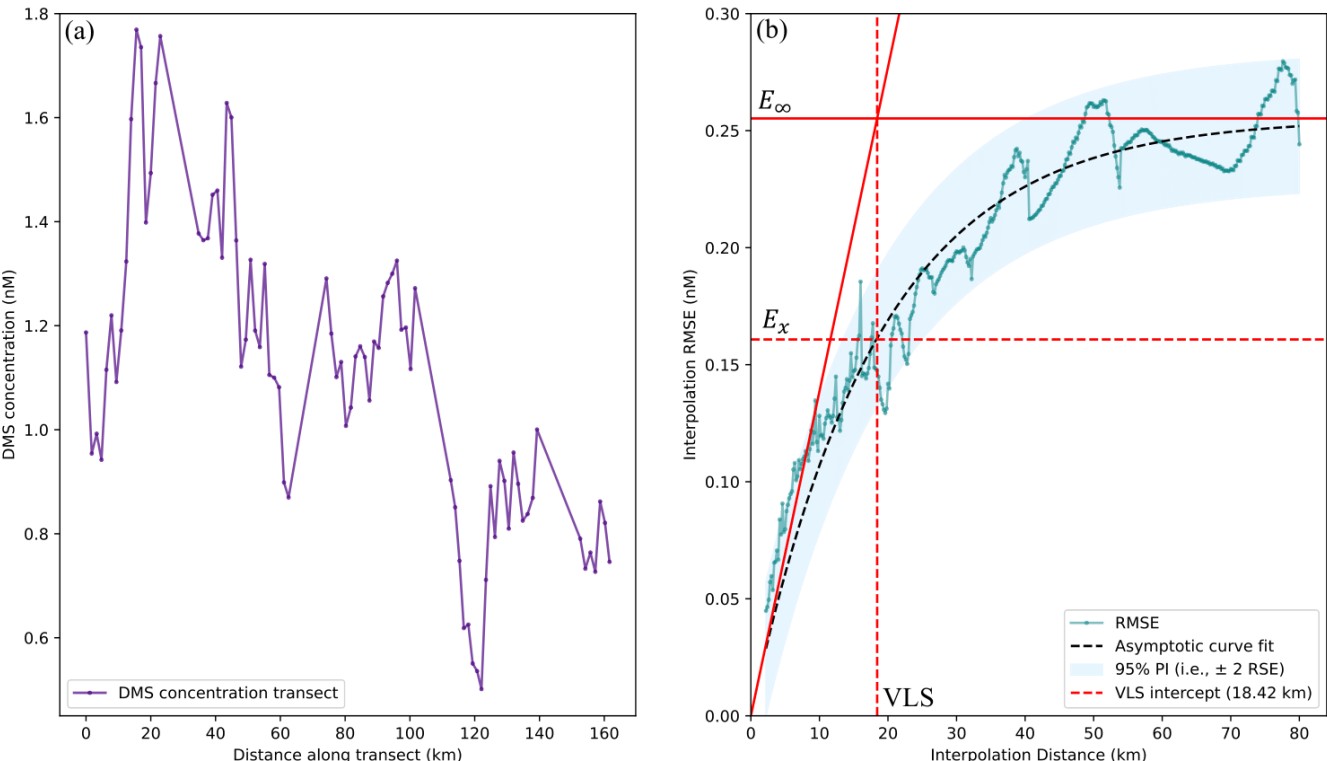

**Figure 2: (a) Example seawater DMS concentration (nM) data transect (sampled from the northwest Atlantic during the NAAMES1**
**(November 2015) campaign; see Bell et al., 2021), analysed to find the variability lengthscale (VLS). (b) Asymptotic error curve**
**(dashed black) fitted to interpolation errors (RMSE, nM; dotted cyan) plotted as a function of increasingly coarse interpolation**
**distance (km). The 95% prediction intervals (PI) of the non-linear regression fit, i.e., ± 2 × residual standard errors (RSE), are**
**shaded blue. The VLS (km) is characterised as the intercept (dashed red) on the curve at 63% of the asymptotically approached**
**maximum interpolation error (nM). Method adapted from Hales & Takahashi (2004).**



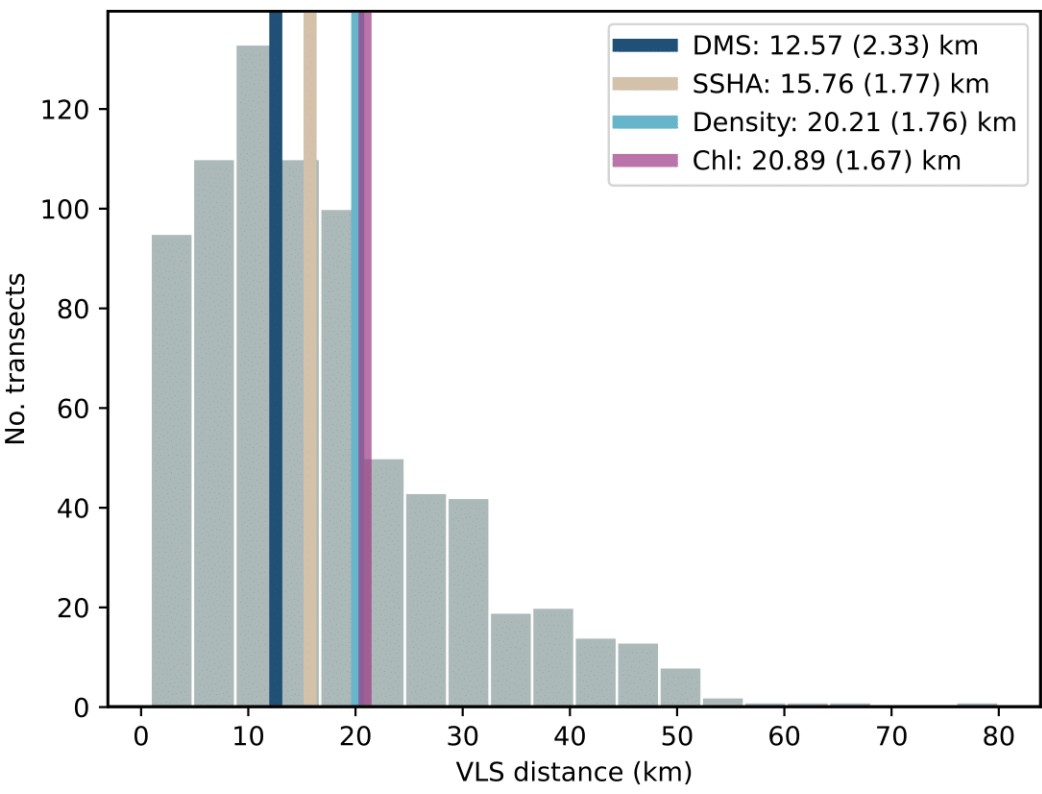


**Figure 3:** Frequency distribution of variability lengthscales (VLS, km) for all DMS transects (grey bars). Vertical coloured lines correspond to the global geometric mean (and geometric standard deviation, GSD) from all transects for VLS$_{DMS}$ (dark blue), VLS$_{SSHA}$ (beige), VLS$_{density}$ (light blue), VLS$_{Chl}$ (pink).


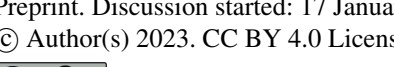



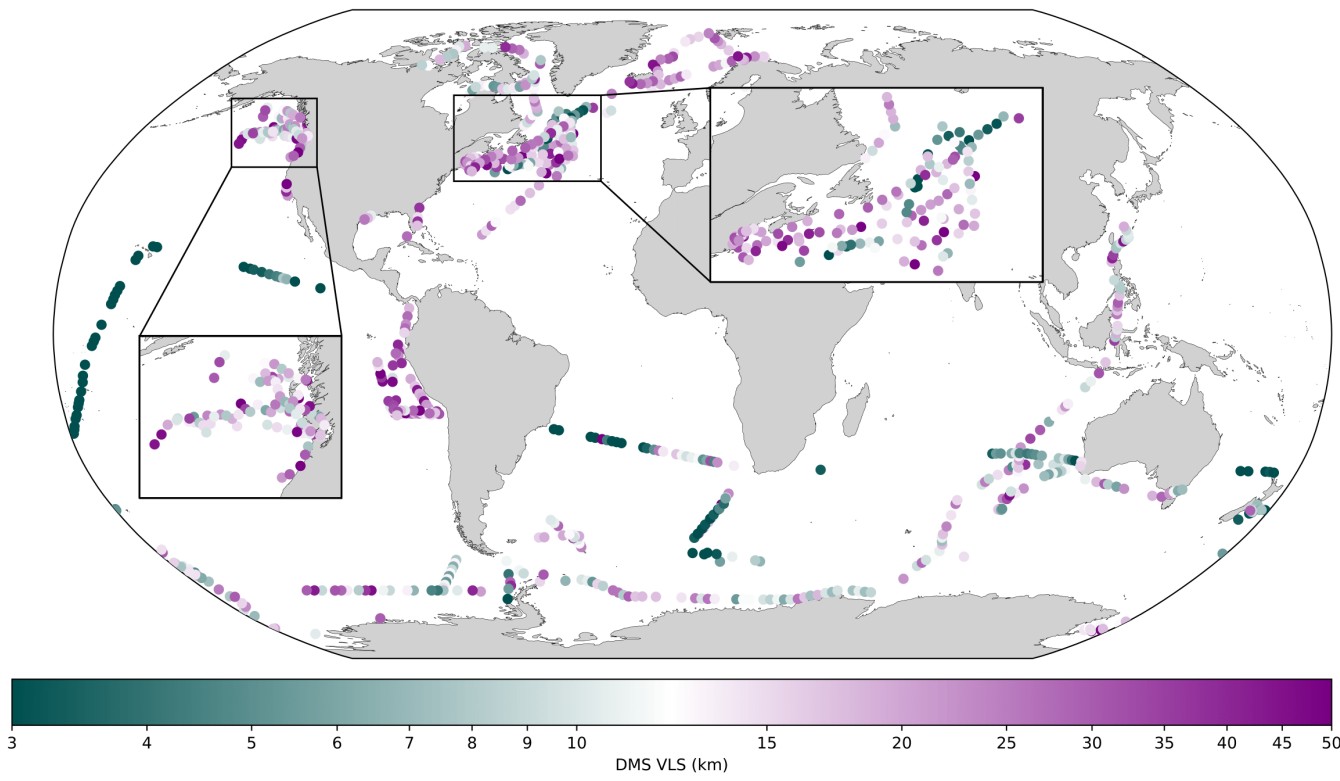

**Figure 4: Global distribution of 763 transects coloured by VLS_DMS (km, log scale). See Fig. S3 in the Supplementary Material for equivalent VLS distribution maps of Chl, density, SSHA, salinity and SST, and Fig. S1 for the spatiotemporal distribution of**
**VLS_DMS.**





**Figure 5: Campaign average VLS$_{DMS}$ (km) plotted versus (a) VLS$_{density}$, (b) VLS$_{SSHA}$, and (c) VLS$_{Chl}$. Error bars indicate 1 GSD of the data within each campaign.**








**Table 1: Regression results for the prediction of campaign average VLS$_{DMS}$, using different combinations of input parameters. Models are ranked in order of how much VLS$_{DMS}$ variance is explained. Models that are significant ($p < 0.01$) are denoted using \*.**

| Model no. | Input parameters | $R^2$ | Adj. $R^2$ | p | Relative importance (%) | N (no. of campaigns) | No. transects used to calculate campaign averages (of 760) |
|---|---|---|---|---|---|---|---|
| Linear Regression | | | | | | | |
| 1 | VLS$_{Chl}$ | 0.47 | – | <0.01* | 100 | 29 | 351 |
| 2 | VLS$_{SSHA}$ | 0.46 | – | <0.01* | 100 | 24 | 361 |
| 3 | VLS$_{density}$ | 0.37 | – | <0.01* | 100 | 32 | 480 |
| 4 | VLS$_{salinity}$ | 0.33 | – | <0.01* | 100 | 32 | 490 |
| 5 | VLS$_{SST}$ | 0.21 | – | 0.014 | 100 | 28 | 445 |
| 6 | Latitude (abs.) | 0.02 | – | 0.375 | 100 | 35 | 760 |
| Multiple Linear Regression | | | | | | | |
| 7 | VLS$_{Chl}$ VLS$_{SSHA}$ VLS$_{density}$ | 0.83 | 0.77 | <0.01* | 34 52 14 | 12 | 87 |
| 8 | VLS$_{Chl}$ VLS$_{SSHA}$ VLS$_{salinity}$ | 0.77 | 0.71 | <0.01* | 58 41 1 | 15 | 100 |
| 9 | VLS$_{Chl}$ VLS$_{density}$ | 0.66 | 0.63 | <0.01* | 54 46 | 26 | 224 |
| 10 | VLS$_{salinity}$ VLS$_{SST}$ | 0.62 | 0.59 | <0.01* | 85 15 | 25 | 322 |
| 11 | VLS$_{Chl}$ VLS$_{SSHA}$ | 0.51 | 0.46 | <0.01* | 35 65 | 20 | 177 |
| 12 | VLS$_{Chl}$ VLS$_{salinity}$ | 0.50 | 0.45 | <0.01* | 70 30 | 22 | 211 |
| 13 | VLS$_{SSHA}$ VLS$_{salinity}$ | 0.49 | 0.44 | <0.01* | 91 9 | 23 | 234 |
| 14 | VLS$_{Chl}$ VLS$_{SST}$ | 0.46 | 0.4 | <0.01* | 73 27 | 22 | 204 |
| 15 | VLS$_{SSHA}$ VLS$_{SST}$ | 0.43 | 0.36 | <0.01* | 75 25 | 20 | 189 |
| 16 | VLS$_{SSHA}$ VLS$_{density}$ | 0.41 | 0.35 | <0.01* | 84 16 | 22 | 213 |
| 17 | VLS$_{Chl}$ VLS$_{SSHA}$ VLS$_{SST}$ | 0.50 | 0.29 | 0.156 | 84 15 1 | 11 | 77 |