# Peer review of "Global analysis of the controls on seawater dimethylsulfide spatial variability"

_Biogeosciences, 2022_

## Referee Comment (RC1)

Review of Manville et al. Global analysis of the controls on seawater dimethysulfide spatial variability

This article represents a large-scale analysis of spatial DMS variability in the worlds oceans, building on a growing data base of high frequency measurements and applying a standardized methodology to examine characteristic length scales of variability. The analysis demonstrates a significant range in DMS variability length scales, and a statistically significant relationship between the variability length scales of DMS, sea surface height anomalies and chlorophyll. The paper is notable for its broad-based analysis of a large data set, and I think the results are helpful in characterizing DMS spatial variability across various oceanographic regimes in the context of other key environmental variables. Overall the paper is well written and the analysis seems sound. I do have a number of specific suggestions, which I feel would further improve the manuscript.

General comments

Specific comments:

Line 33 add 'loads' after 'aerosol'

Line 36 aerosol should be plural

Line 38. I found the last sentence of this paragraph a bit long and convoluted. I would shorten it, or split into two for greater clarity.

line 41. Add a comma after complex.

Line 60, I think the authors should cite the work of Herr et al., who designed an empirical algorithm for the NE Pacific.

Line 60 – I think the Wange et al. 2020 paper is a global ML climatology. I think it's worth citing the two recent ML-based algorithm by McNabb – one for the S. Ocean, and one for the NE Pacific.

Line 66. At this point, I think there have actually been a pretty significant number of studies looking at sub-mesoscale variability – the authors can pick a few of their favorites and cite them here.

Line 66. Here and elsewhere, I 'm not sure that the authors are really examining 'processes' governing DMS variability. Rather, they are looking for statistical association and speculating on potential mechanisms. I think the distinction is perhaps subtle, but important.

Line 70.  The question of appropriate interpolation radii is key to all of the previous climatologies.  I think this concept should be introduced earlier, when discussing existing algorithms, and their challenges in resolving fine-scale variability.

Line 78.  I'm not sure I'd say there is a 'wealth' of high frequency data – maybe a 'growing number of high frequency DMS measurements' is more accurate.

Line 103.  I would have thought that sampling distance, rather than time would be a better cut-off.  Based on the data used, can the authors list the maximum sampling interval included in the data?  I think it appears later, but would be good here.

Line 105.  I think it would be better to cite the earlier papers that introduced mass spectrometry in the early 2000s.

Line 111.  Again, I'm not sure that the analysis presented here allows the authors to explore 'processes'.

L117-118 – Is there a reason PAR wasn't included, or even the diffuse attenuation coefficient (kd), to assess light variability? Aqua MODIS has 4 km products for each variable that could be compared with the current suite of variables. This would be highly valuable, in my opinion, and would strengthen the discussion points in L326-327 & L370-375.

Line 118.  I presume that the MODIS chl were matched to the month and year corresponding to the DMS measurements, but it would be good to state this explicitly.   I realize that lower temporal averaging increased cloud cover data losses, but one month is a long time for DMS to change.  It is at least worth mentioning the potential temporal disconnect between in situ DMS measurements (which can change significantly over even just a few days) to a monthly-averaged Chl product.  Also, how were the differing spatial resolutions of satellite SST, Chl and SSHA, in relation to DMS, handled? Were the closest matching pixels chosen as "coincident", or were these variables upsampled (interpolated) to a finer resolution to match?

Moreover, the mean VLSSSHA of 15.76 is lower than the resolution of the satellite product used (~18 km). If the SSHA data was not upscaled, shouldn't the minimum VLS determined be constrained by the raw sampling resolution (i.e. VLS shouldn't be able to detect variability within a single, averaged grid cell/pixel)? If the VLSSSHA is more closely representing an approximation, than a caveat should be noted.

General methods question.  What is the potential advantage / disadvantage of linearly-detrended the data prior to VLS analysis?  I don't think this was done here, and the data in Fig. 2a certainly a show a strong linear trend.  What is the impact of this on the analysis?

First couple of paragraphs of the results.  I think some more statistical analysis is warranted to examine significant differences between different values mentioned.  I would also suggest representing mean and GSD values at xx ± yy.

Section 3.2

I found the first paragraph a bit 'jumbly', as it moved quickly across very different regimes. In general, I didn't think the spatial analysis was all that clear or convincing. I think the statement about 'consistently' small VLSdms in the south tropical gyres has some notable exceptions which seem to be glossed over (same comment for line 266). Line 210 – it would be good to report the actual values for different areas (e.g. East Equatorial Pacific).

Line 231. The highest explanatory power comes from using SSHA and Chl as predictive variables. So I don't understand why all statistically significance is lost when SST is added. To my understanding, you would simply fail to get a lower r2 in the MLR when adding an extra variable with no correlation to the dependent variable.

Line 261 – 'broadly above average' seems a bit vague to me.

Line 262 'invariability' seems a bit awkward – maybe 'the longer length scales of DMS spatial variability ….'

Line 275 Herr et al. (2019) explicitly link DMS variability with SSHA and eddys. See figure copied below

[Figure]

**Figure 5.** Line plot of sea surface height anomaly (SSHA) on 15 July 2016 and observed DMS concentrations between 14 and 16 July 2016 along T1. DMS along the T1 transect is highest in those areas influenced by positive SSHA values.

Line 288. Are the VLSs of Chl and SSHA correlated to each other? If so, is that a problem in the analysis, creating a potential statistical artifact?

Line 294. I think there should be more explicit discussion of Rossby radii here as a structuring mechanism for different length scales of variability. Some of this material comes up later in the text (lines 346 – 350), but I think it would be good here.

Section 4.4. I'm not sure what this title is supposed to mean – I didn't find it too descriptive.

L321-323 – This argument could be extended using this study's results. The VLS for all variables is about 20 km or less, but current regression-based empirical algorithms (cited in L322) have been built using predictor data interpolated to 1o (111 km) or courser resolution. The VLS results do support the choice of predictors used in these studies, but they also suggest patterns associated with mesoscale variability (particularly associated with SSHA) would be obscured at

those resolutions, motivating modelling work at finer resolutions (e.g. McNabb & Tortell 2022 cited).

L326-327 - This is partially true, but they haven't assessed light variability which is parameterized in all three algorithms cited in L322. The sentence structure also needs revising here.

Line 378 – first line of conclusions. I think the observations are regional and the data set is global. I would re-write this to clarify.

Fig. 4 (& relevant to S1, S3) – It looks like the colorbar diverges at the global geometric average VLSDMS. It might be helpful to make a note of this in the caption to draw the reader's attention.

Figure 5. I would suggest adding another panel to show the relationship between predicted (from the best MLR model) and observed VLSdms.

END OF REVIEW

---

## Author Comment (AC1)

**Author response to reviewers' comments on bg-2022-250**

Comments by reviewers are shown in an italic typeface and the responses are shown in a normal typeface. Corresponding changes have been made and tracked in the revised manuscript and supplement, and are marked in red.

We would like to thank the editor and anonymous reviewers for taking the time to provide such constructive feedback. Their insightful suggestions have contributed to meaningful improvements in the manuscript.
* * *
**RC1: 'Comment on bg-2022-250', Anonymous Referee #1, 13 Feb 2023**

*Review of Manville et al. Global analysis of the controls on seawater dimethylsulfide spatial variability*
*This article represents a large-scale analysis of spatial DMS variability in the worlds oceans, building on a growing data base of high frequency measurements and applying a standardized methodology to examine characteristic length scales of variability. The analysis demonstrates a significant range in DMS variability length scales, and a statistically significant relationship between the variability length scales of DMS, sea surface height anomalies and chlorophyll. The paper is notable for its broad-based analysis of a large data set, and I think the results are helpful in characterizing DMS spatial variability across various oceanographic regimes in the context of other key environmental variables. Overall the paper is well written and the analysis seems sound. I do have a number of specific suggestions, which I feel would further improve the manuscript.*

*General comments*

*Specific comments:*

- *Line 33 add 'loads' after 'aerosol'*

**Response:** We agree, change made.

- *Line 36 aerosol should be plural*

**Response:** We agree, change made.

- *Line 38. I found the last sentence of this paragraph a bit long and convoluted. I would shorten it, or split into two for greater clarity.*

**Response:** We agree, change made to separate into two sentences which now read:

"The distribution of natural marine aerosol sources should be represented at the resolution required to capture the frequency and magnitude of their variability. This is critical for reducing the large uncertainties associated with natural aerosol-cloud interactions."

- *Line 41. Add a comma after complex.*

**Response:** We agree, change made.

- *Line 60, I think the authors should cite the work of Herr et al., who designed an empirical algorithm for the NE Pacific.*

**Response:** We agree, change made.

- *Line 60 – I think the Wange et al. 2020 paper is a global ML climatology. I think it's worth citing the two recent ML-based algorithm by McNabb – one for the S. Ocean, and one for the NE Pacific.*

**Response:** Change made to include the recent two McNabb and Tortell studies, and the Humphries et al. 2012 study. The sentence now reads:

"More recently, global and regional climatologies have been generated using machine learning approaches (Humphries et al., 2012; McNabb & Tortell, 2022, 2023; Wang et al., 2020)."

- *Line 66. At this point, I think there have actually been a pretty significant number of studies looking at sub-mesoscale variability – the authors can pick a few of their favorites and cite them here.*

**Response:** We agree, this sentence has been changed and the paragraph will now start with:

"Recent studies have focussed on local and sub-mesoscale DMS variability, taking advantage of improvements to seawater DMS concentration sampling resolution (e.g., Asher et al., 2011; Nemcek et al., 2008; Tortell, 2005a, 2005b; Tortell & Long, 2009; Zindler et al., 2014)."

- *Line 66. Here and elsewhere, I 'm not sure that the authors are really examining 'processes' governing DMS variability. Rather, they are looking for statistical association and speculating on potential mechanisms. I think the distinction is perhaps subtle, but important.*

Response: We agree and have changed 'processes' to 'potential mechanisms' wherever relevant throughout.

- *Line 70. The question of appropriate interpolation radii is key to all of the previous climatologies. I think this concept should be introduced earlier, when discussing existing algorithms, and their challenges in resolving fine-scale variability.*

**Response:** The following sentence has been added when discussing existing algorithms to highlight the differences/uncertainties in climatologies due to smoothing techniques:

"Significant differences in these smoothed climatological estimates, and thus uncertainties, have been attributed to the gap filling techniques used, specifically the appropriate interpolation/smoothing radius of influence (Hulswar et al., 2022)."

- *Line 78. I'm not sure I'd say there is a 'wealth' of high frequency data – maybe a 'growing number of high frequency DMS measurements' is more accurate.*

**Response:** We agree, change made.

- *Line 103. I would have thought that sampling distance, rather than time would be a better cut-off. Based on the data used, can the authors list the maximum sampling interval included in the data? I think it appears later, but would be good here.*

**Response:** Data from the PMEL DMS database (date of last access: 15 April 2022) were initially screened by sampling frequency using the time between measurements and the number of measurements. Time between measurements and ship speeds were then used to calculate distances between measurements, which were later used to separate data into continuous transects with a minimum cumulative length. We have now introduced the maximum sampling distance earlier, in the first paragraph of Section 2.1. The sentence reads:

"Selection criteria are used to identify datasets suitable for sub-mesoscale VLS analysis: a minimum of 100 data points in total and ≤1 hour between measurements, which excludes all data with a spatial resolution >30 km."

- *Line 105. I think it would be better to cite the earlier papers that introduced mass spectrometry in the early 2000s.*

**Response:** We agree, change made.

- *Line 111. Again, I'm not sure that the analysis presented here allows the authors to explore 'processes'.*

**Response:** We have changed 'processes' to 'potential mechanisms'.

- *L117-118 – Is there a reason PAR wasn't included, or even the diffuse attenuation coefficient (kd), to assess light variability? Aqua MODIS has 4 km products for each variable that could be compared with the current suite of variables. This would be highly valuable, in my opinion, and would strengthen the discussion points in L326-327 & L370-375.*

**Response:** This matches a similar comment made by anonymous reviewer #2 with regards to why we chose certain variables. We agree, this is an excellent point, and a measure of in-situ light variability would have been extremely valuable and strengthened our analysis. Unfortunately, PAR at the ocean surface is much more variable than sea surface height anomalies and chlorophyll, which can be averaged at lower resolution to maximise coverage and minimize cloud

coverage data losses. Analysing the response of ocean parameters to variability in light experienced at the surface would require light at the surface to be accurately measured in all (both cloudy and clear sky) conditions. Additionally, plankton light exposure is not controlled by ocean surface PAR alone, but by the 'solar radiation dose' in the actively mixing layer which depends on surface irradiance, mixing layer depth ('XLD'), Kd, etc. Combining satellite PAR and Kd with climatological MLD (not XLD) may not capture actual light exposure, as briefly discussed in Galí et al. (2018). Ideally, we would have utilised in-situ high frequency surface PAR measurements coincident to the DMS measurements, but this was typically not available, so we chose not to include PAR as a parameter in our analysis.

- *Line 118. I presume that the MODIS chl were matched to the month and year corresponding to the DMS measurements, but it would be good to state this explicitly. I realize that lower temporal averaging increased cloud cover data losses, but one month is a long time for DMS to change. It is at least worth mentioning the potential temporal disconnect between in situ DMS measurements (which can change significantly over even just a few days) to a monthly-averaged Chl product. Also, how were the differing spatial resolutions of satellite SST, Chl and SSHA, in relation to DMS, handled? Were the closest matching pixels chosen as "coincident", or were these variables upsampled (interpolated) to a finer resolution to match?*

**Response:** We agree that the details of satellite data processing could be clearer and so the opening sentences to the paragraph has been modified to reflect this, as follows:

"Satellite monthly mean chlorophyll-a (Chl) and 5-day sea surface height anomaly (SSHA) data are matched to the average date of each DMS sampling cruise. Satellite data pixels are extracted along the coordinates of the DMS cruise track using the NASA SeaDAS software (version 7.5.3)."

Chlorophyll and sea surface height anomaly data were extracted at the closest matching pixels to the DMS coordinates, they were not interpolated. The limitations due to different satellite and DMS sampling resolutions are discussed in Section 4.5, paragraph 2. SST and salinity (and therefore density) data are in-situ and sampled at the same resolution as the DMS data.

- *Moreover, the mean VLSSSHA of 15.76 is lower than the resolution of the satellite product used (~18 km). If the SSHA data was not upscaled, shouldn't the minimum VLS determined be constrained by the raw sampling resolution (i.e. VLS shouldn't be able to detect variability within a single, averaged grid cell/pixel)? If the VLSSSHA is more closely representing an approximation, than a caveat should be noted.*

**Response:** The SSHA data resolution is 0.17°, which only equates to a maximum of ~18km at the equator. Poleward of ~30° the global average $VLS_{SSHA}$ is higher than the SSHA resolution. As discussed in Section 4.5, only seven of the 37 campaigns in this study are in low latitude (<30°) waters. Again, the limitations of the data resolutions are discussed in Section 4.5.

We fit an asymptotic curve to the RMSE of each parameter as a function of increasing sub-sampling distance. The curve is used to ascertain the variability lengthscale if it meets the quality control criteria i.e., it describes the data sufficiently well. If the curve is considered to fit the data well, we can have confidence that the VLS can be objectively identified using the curve, even if this falls below the sampling frequency.

- *General methods question. What is the potential advantage / disadvantage of linearly-detrended the data prior to VLS analysis? I don't think this was done here, and the data in Fig. 2a certainly a show a strong linear trend. What is the impact of this on the analysis?*

**Response:** Detrending the data would remove an underlying low frequency mode of variability. The aim and 'real-world' application of VLS analysis is to identify and inform the lowest sampling resolution necessary to capture most of the DMS variability in the ocean. By removing low frequency of variability prior to performing VLS analysis, we would risk losing information about the dominant mode of variability in the data. In our study, 276 out of 1039 DMS transects were filtered out at the quality control stage. This indicates that the dominant scale of variability in these transects could not be identified between 50-99.5 km i.e., with transects 100-200 km in length. For reasons discussed in Section 4.5, we use consistent transect lengths for all data in our study. The downside to this choice is that it is harder to pick out variability associated with large physical features, especially at low latitudes.

- *First couple of paragraphs of the results. I think some more statistical analysis is warranted to examine significant differences between different values mentioned. I would also suggest representing mean and GSD values at xx ± yy.*

**Response:** We agree with the first suggestion and have provided the Mann-Whitney $U$ Test statistical significance ($p$ <0.01) of the difference between global average VLS values. It is our understanding that GSD is a dimensionless multiplicative factor, i.e., a GSD of 2 means points are typically spread within a factor of 2 from the geometric mean. The GSD values have been provided only as parametric test to indicate the spread of the data from the central tendency.

*Section 3.2*

- *I found the first paragraph a bit 'jumbly', as it moved quickly across very different regimes. In general, I didn't think the spatial analysis was all that clear or convincing. I think the statement about 'consistently' small VLSdms in the south tropical gyres has some notable exceptions which seem to be glossed over (same comment for line 266). Line 210 – it would be good to report the actual values for different areas (e.g. East Equatorial Pacific).*

**Response:** We have modified the section to improve the structure, remove 'consistently', and include notable exceptions to the small $VLS_{DMS}$. We now refer the reader to campaign average $VLS_{DMS}$ for actual values in different areas which are compiled in the Supplementary Material, Table S2. We also provide average $VLS_{DMS}$ for the M10 circumnavigation, to highlight the contrast with large $VLS_{DMS}$ in the east equatorial Pacific. The section now reads:

"VLS$_{DMS}$ is generally small in the subtropical gyres, specifically the equatorial and subtropical South Pacific and South Atlantic (Fig. 4; see Table S2 for each sampling campaign average VLS$_{DMS}$, e.g., campaign numbers: 30, M10a, black; 31, M10b, dark red; and 32, M10c, cyan). The average VLS$_{DMS}$ from all transects in the three M10 low–mid latitude circumnavigation campaign datasets (mean = 6.34 km, GSD = 2.59) is consistently smaller than the global value (mean = 12.57 km, GSD = 2.33) (Fig. 4). The relative homogeneity of small VLS$_{DMS}$ in these oligotrophic domains is not replicated in the VLS of any other variables (Fig. S3). The Southern Hemisphere subtropical gyres are permanently stratified biomes, bounded to the south by a band of seasonally stratified biomes (Fay & McKinley, 2014). At the boundary transitions from permanently to seasonally stratified conditions there are some notable exceptions to the low VLS$_{DMS}$, e.g., the Benguela upwelling (southeast Atlantic) and South Australia upwelling (Fig. 4).

In contrast, the average (mean = 22.06 km, GSD = 1.60) of VLS$_{DMS}$ in the Peruvian upwelling (East equatorial Pacific) is consistently larger than the global average (mean = 12.57 km, GSD = 2.33) (Fig. 4). Larger VLS$_{DMS}$ is also found along parts of the Pacific and Atlantic coastlines of North America, with smaller VLS$_{DMS}$ further offshore (Fig. 4, inset). VLS$_{DMS}$ in the Arctic, northeast Pacific, northwest Atlantic, and southeast Indian open ocean regions are highly variable. The Southern Ocean has VLS$_{DMS}$ generally below the global average and features some localised pockets with larger VLS (Fig. 4). DMS concentration variability in mid–high latitude regions is seasonal (Hulswar et al., 2022) and VLS$_{DMS}$ could be influenced by the season / time of year."

- *Line 231. The highest explanatory power comes from using SSHA and Chl as predictive variables. So I don't understand why all statistically significance is lost when SST is added. To my understanding, you would simply fail to get a lower r2 in the MLR when adding an extra variable with no correlation to the dependent variable.*

**Response:** The $R^2$ is almost unchanged from model 11 to model 17 i.e., with the addition of SST, which as you say is to be expected. However, this is where the adjusted $R^2$ is valuable and demonstrates why it has been used throughout the study in place of $R^2$, for correlations involving more than one independent variable. The adjusted $R^2$ considers the number of independent variables and shows that including SST in model 17 does not improve the model fit.

It is also important to note that there is a substantial loss of data between model 11 and model 17 due to the inclusion of SST as a third independent variable. This is because prior to correlation, the input data are filtered to only include data transects where all input parameters are co-located, as discussed in Section 3.3.2. Therefore, the VLS$_{SSHA}$ and VLS$_{Chl}$ data used to predict VLS$_{DMS}$ in model 17 are a smaller subset of those used in model 17. See Table S3, Supplementary Material for regression results without filtering for input data co-location.

- *Line 261 – 'broadly above average' seems a bit vague to me.*
**Response:** We have changed 'broadly' to 'generally'.

- *Line 262 'invariability' seems a bit awkward – maybe 'the longer length scales of DMS spatial variability ....'*

**Response:** We agree, change made.

- *Line 275 Herr et al. (2019) explicitly link DMS variability with SSHA and eddys. See figure copied below*

[Figure]

**Figure 5.** Line plot of sea surface height anomaly (SSHA) on 15 July 2016 and observed DMS concentrations between 14 and 16 July 2016 along T1. DMS along the T1 transect is highest in those areas influenced by positive SSHA values.

**Response:** We agree, change made to include this citation.

- *Line 288. Are the VLSs of Chl and SSHA correlated to each other? If so, is that a problem in the analysis, creating a potential statistical artifact?*

**Response:** It is our understanding that multicollinearity (two or more related predictor variables in a multiple linear regression) doesn't influence the model's predictive power, reliability, or the goodness-of-fit. From Kutner et al. 2005 *Applied Linear Statistical Models*: "The fact that some or all predictor variables are correlated among themselves does not, in general, inhibit our ability to obtain a good fit nor does it tend to affect inferences about mean responses or predictions of new observations". As described earlier, our use of the adjusted $R^2$ for each model accounts for the number of predictor variables used, and the dominance analysis is then independently used to determine the incremental $R^2$ contribution of each predictor to the complete model. Using these statistical methods we can have confidence in the predictive power of each multiple linear regression model, and the relative contribution of the predictor variables.

- *Line 294. I think there should be more explicit discussion of Rossby radii here as a structuring mechanism for different length scales of variability. Some of this material comes up later in the text (lines 346 – 350), but I think it would be good here.*

**Response:** We agree, change made to introduce the latitudinal dependence of Earth's rotation effects (Coriolis, Rossby radius) and its impact on lengthscales of variability. We also highlight the difference in transect lengths used by Royer et al. 2015 and point the reader to Section 4.5 where the implications of this are discussed in more detail. The sentences now read:

"These contrasting conclusions potentially reflect the fact that length scales of physical oceanographic variability increase towards the equator, due to the effects of the Earth's rotation. The Coriolis parameter and therefore Rossby radius are intrinsically latitudinal dependent (Jacobs et al., 2001). The longer transects used by Royer et al. (2015) at low–mid latitudes enable them to capture scales of variability that may be associated with large physical features. This point is discussed further in in Section 4.5."

- *Section 4.4. I'm not sure what this title is supposed to mean – I didn't find it too descriptive.*

**Response:** We agree, change made. The title now reads:

"Implications for global DMS parameterisation"

We point the reviewer to all of Section 4.4, which has been amended and restructured in response to comments by both reviewers #1 & #2.

- *L321-323 – This argument could be extended using this study's results. The VLS for all variables is about 20 km or less, but current regression-based empirical algorithms (cited in L322) have been built using predictor data interpolated to 1o (111 km) or courser resolution. The VLS results do support the choice of predictors used in these studies, but they also suggest patterns associated with mesoscale variability (particularly associated with SSHA) would be obscured at those resolutions, motivating modelling work at finer resolutions (e.g. McNabb & Tortell 2022 cited).*

**Response:** We thank the reviewer for their interpretation here and have included this point in our discussion. The sentence now reads:

"Our results indicate that patterns of mesoscale and sub-mesoscale DMS variability, particularly those associated with SSHA, will be obscured at the 1° resolution of most global parameterisations, highlighting the importance of modelling work at finer resolutions (e.g., Galí et al., 2019; McNabb & Tortell, 2022, 2023)."

- *L326-327 - This is partially true, but they haven't assessed light variability which is parameterized in all three algorithms cited in L322. The sentence structure also needs revising here.*

**Response:** We agree, the sentence now includes 'key' variables – highlighting that not all parameterised variables are included – and has been restructured as follows:

"This study supports the choice of key variables used in existing empirical parameterisations by demonstrating that, even on small scales, physical mixing (SSHA, density) and biological activity (Chl) explain a large portion of surface seawater DMS spatial variability in the global ocean."

- *Line 378 – first line of conclusions. I think the observations are regional and the data set is global. I would re-write this to clarify.*

**Response:** We agree, change made. The sentence now reads:

"This study presents a comprehensive and objective analysis of DMS variability based on a large global dataset of high frequency observations at the local/regional scale."

- *Fig. 4 (& relevant to S1, S3) – It looks like the colorbar diverges at the global geometric average VLSDMS. It might be helpful to make a note of this in the caption to draw the reader's attention.*

**Response:** We agree, change made to Figure 4, S1 & S3.

- *Figure 5. I would suggest adding another panel to show the relationship between predicted (from the best MLR model) and observed VLSdms.*

**Response:** We agree, change made to include the predicted $VLS_{DMS}$ panel, which is now discussed briefly at the end of Section 4.3.

*END OF REVIEW #1*

**RC2: 'Comment on bg-2022-250', Anonymous Referee #2, 16 Feb 2023**

*This manuscript uses variability length scales to characterise dimethylsulfide (DMS) distributions in the global ocean. There have been many attempts to parametrise/predict DMS concentrations, but often they are insufficient for describing either specific events or global values. DMS in the ocean is controlled by many biotic and abiotic factors, which makes the task difficult. However, since sulfur via DMS emitted to the atmosphere is abundant and plays an important role in aerosol and cloud formation over the ocean, there is great interest in understanding DMS distributions in space and time. It is also of great interest to understand how dissolved DMS and its subsequent flux to the atmosphere are altered through climate/environmental change. The method outlined here seems to be robust and relatively universal – can be applied to global measurements. Thefinding that there is a characteristic variability length scale range for DMS is interesting and useful. The manuscript is generally well written and clear and should be accepted for publication after the following comments have been sufficiently addressed.*

*Specific comments:*

- *General – all data from the PMEL database that is used should be referenced (if there are references in the database). If there are no references, then the acknowledgement (as has been done already by the authors) is appropriate. This is standard practice (e.g., SOCAT data policy, which can be found here: https://www.socat.info/wp-content/uploads/2020/06/Data_Use_Statement-for-SOCAT_v2020.pdf).*

**Response:** We agree, change made by adding all DMS data references to Table S2 in the Supplementary Material.

- *Line 53 - reference typo*

**Response:** We agree, change made.

- *Line 67 - first sentence should reference works that are appropriate, as it says few studies (implying there have been studies). Perhaps appropriate is Zindler et al. (2014, https://doi.org/10.1002/2014GL059547) as it uses high resolution data and investigates the role of eddies (although not sub-mesoscale, I suppose, but regional). Also, was an effort made to check the literature for appropriate datasets that were not yet input to the PMEL database? Seems like there are several more that could be used.*

**Response:** This comment agrees with a similar comment made by anonymous reviewer #1. We agree and have changed the sentence, so it now reads as follows:

"Recent studies have focussed on local and sub-mesoscale DMS variability, taking advantage of improvements to seawater DMS concentration sampling resolution (e.g., Asher et al., 2011; Nemcek et al., 2008; Tortell, 2005a, 2005b; Tortell & Long, 2009; Zindler et al., 2014)."

With regards to the second point, an effort was duly made to incorporate DMS data that were not in the PMEL database, hence the inclusion of data from the Malaspina Expedition, NAAMES cruises, and the SCALE project in this study (see Table S2, Supplementary Material for details). This was done as part of a wider effort to bring together all available DMS data (and utilise VLS analysis) in the recently released DMS-Rev3 observational climatology (Hulswar et al. 2022), for which many of us were involved as authors.

- *Line 105 – Perhaps the Saltzman et al. (2009, 10.5194/os-5-537-2009) paper should also be referenced here?*
**Response:** We agree, change made.

- *Line 163 – It might be useful to give a brief explanation as to why only these parameters were chosen for analysis. This point comes up again in the section on limitations. It might be useful in that section to learn some way to incorporate important missing parameters in this type of analysis in the future.*
**Response:** This is an excellent point and matches a similar one made by reviewer #1 with regards to our study lacking the inclusion of PAR. We made use of all available high frequency in-situ SST, salinity and derived density data, which most reliably capture the physical oceanographic conditions in which the DMS measurements were made. Unfortunately, in-situ sea surface height anomaly (SSHA), biological productivity/biomass (proxied using chlorophyll) and light data were not available coincident to the DMS measurements included in our study. SSHA and chlorophyll can be reasonably aggregated to lower temporal resolutions to maximise coverage, the caveats of which are discussed in Section 4.5. On the other hand, light experienced at the surface depends on incoming shortwave irradiance, the diffuse attenuation coefficient, and the actively mixing layer. It is hard to capture fine scale light exposure from satellite data (or reanalysis) and with MLD which is generally climatological, as briefly discussed in Galí et al. (2018). As such, we chose not to include light in our analysis. We certainly recommend future studies make use of in-situ light and chlorophyll measurements where possible and would urge the observational communities to collaborate closely in compiling combined datasets with coincident measurements of related parameters.

- *Section 4.4 – This paragraph is a bit unclear to me. I do not understand why the first sentence is used. It seems completely disconnected from the next few sentences. Also, the sentence starting on line 320 is not so clear – what is the powerful tool provided? I think this is just a wording problem. Finally, the last two sentences make wishy-washy arguments (is this why there is a question mark in the subheading?). I guess it is no big surprise that physical mixing and biological activity drive DMS distributions and to say that DMS spatial variability is partially explained by these*

*factors seems too simple (and a bit like we are starting here from zero – the DMS community knows the general drivers of DMS distributions, but can't predict them). The next sentence says that the significant correlations presented here were again found by averaging – which is exactly the problem underscored on line 323. Is there an attempt here at solving something? Are the authors trying to say that the parametrisations can be improved by find the right combination of parameters exhibiting sub-mesoscale variability? Or is there some other point to this paragraph?*

**Response:** We have taken the comments from reviewers #1 & #2 together and revised Section 4.4. Changes include the title, the first sentence, and the removal of 'powerful tool' in response to the comments by reviewer #2. The point made by reviewer #1 to extend the argument from L321-323 has been implemented, and hopefully answers the comment made by reviewer #2 about the last two sentences in the section. The section has been restructured and now reads as follows:

**"4.4 Implications for global DMS parameterisation**

Low resolution measurements have previously been used to predict mean spatiotemporal patterns of DMS, both regionally and globally. Several studies have parameterised DMS as a function of surface  mixed layer depth (MLD), light, and Chl (Anderson et al., 2001; Aranami & Tsunogai, 2004; Aumont et al., 2002; Belviso et al., 2004; Galí et al., 2018; Simó & Dachs, 2002; Vallina & Simó, 2007). For example, Simó & Dachs (2002) use climatological MLD and remotely sensed Chl to estimate average DMS concentrations, while Vallina & Simó (2007) use climatological MLD, surface irradiance and light attenuation to estimate surface DMS from the 'solar radiation dose'. Galí et al. (2018) employ an algorithm driven by climatological Argo MLD and satellite derived Chl, SST, and photosynthetically active radiation (PAR). Only the latter algorithm was additionally validated at finer resolution using non-climatological data to enable regional timeseries studies (Galí et al., 2019). SSHA reflects surface mixing and changes to the MLD (Gaube et al., 2019). This study supports the choice of key variables used in existing empirical parameterisations by inferring that physical mixing (SSHA, density) and biological activity (Chl) explain a large portion of the spatial variability of surface seawater DMS in the global ocean even on small scales.

DMS parameterisations with global coverage that rely on remote and autonomous observations predict spatially and seasonally averaged surface seawater DMS reasonably well (e.g., Galí et al., 2018; Simó & Dachs, 2002; Vallina & Simó, 2007). However, some studies have questioned whether such parameterisations are overly reliant on spatial/temporal averaging, often to 1°/monthly resolution (e.g., Derevianko et al., 2009). The spatiotemporal averaging used to develop global parameterisations may lead to an over-confidence in current predictive capabilities because key parameters are not included. Statistically significant MLR relationships in this study are obtained once the transect data are averaged by campaign, and the average VLS for all six variables in our study is tens of kilometres. Using VLS analysis to assess the covariation of parameters at the sub-mesoscale provides insights that can help to improve global parameterisations. Our results indicate that patterns of mesoscale and sub-mesoscale DMS variability, particularly those associated with SSHA,

will be obscured at the 1° resolution of most global parameterisations, highlighting the importance of modelling work at finer resolutions (e.g., Galí et al., 2019; McNabb & Tortell, 2022, 2023). Regional studies have tested empirical predictive relationships for DMS with varying degrees of success (Asher et al., 2011; Bell et al., 2006, 2021; Royer et al., 2015)."